

# Adaptive robust structure exploration for complex systems based on model configuration and fusion

Yingfei Qu[1], Wanbing Liu[2], Junhao Wen[1] and Ming Li[3]

[1] Computer Science and Technology Post-Doctoral Station, Chongqing University, Chongqing, China
[2] Hengda Fuji Elevator Co. Ltd., Huzhou, China
[3] Chongqing Key Laboratory for Intelligent Perception and Blockchain Technology, Chongqing Technology and Business University, Chongqing, China

## ABSTRACT

Analyzing and obtaining useful information is challenging when facing a new complex system. Traditional methods often focus on specific structural aspects, such as communities, which may overlook the important features and result in biased conclusions. To address this, this article suggests an adaptive algorithm for exploring complex system structures using a generative model. This method calculates and optimizes node parameters, which can reflect the latent structural characteristics of the complex system. The effectiveness and stability of this method have been demonstrated in comparative experiments on 10 sets of benchmark networks using our model parameter configuration scheme. To enhance adaptability, algorithm fusion strategies were also proposed and tested on two real-world networks. The results indicate that the algorithm can uncover multiple structural features, including clustering, overlapping, and local chaining. This adaptive algorithm provides a promising approach for exploring complex system structures.

# INTRODUCTION

With the advancement of digital transformation (*Zaoui & Souissi, 2020*; *Kraus et al., 2021*) complex network systems are experiencing exponential growth (*Cohen et al., 2022*; *Rosas et al., 2022*; *Shurety, Bodin & Cumming, 2022*; *Stella, 2022*). There is a shortage of manpower and resources for researching complex network systems. Therefore, a highly adaptive framework is needed to automatically explore the structures of complex networks without requiring extensive prior knowledge.

Despite having been studied for many years, complex network structure exploration (*Strogatz, 2001*; *Wei, Xu & Ma, 2019*; *Li et al., 2020*) remains an active research field due to its widespread application (*Mou et al., 2020*; *Zhang et al., 2020*) and significant value in solving real-world problems (*Li et al., 2021*; *Lei & Cheong, 2022*; *Zhao et al., 2022*). In a commentary article published in Nature Physics, *Fortunato & Newman (2022)* review the advancements in community detection, a crucial direction in exploring network structures, over the past two decades. The article provides an overview of representative community

Corresponding author
Wanbing Liu, jpmf007@qq.com

detection techniques, identifies the detection limitations encountered in community discovery, and affirms the data processing capabilities of representation learning. The team led by *Hui-Jia et al. (2022)* has developed metrics and models to assess network structure, allowing for a quantitative evaluation of structural exploration. *Xu et al. (2021)* found that the Kuramoto oscillation model's synchronization algorithm could be employed to reveal the overlapping structural features of a network. The approach introduced by *Ma et al. (2021)* involves hierarchical partitioning of networks in network systems based on scale indicators. They further propose a network structure exploration algorithm based on joint non-negative matrix factorization, allowing for structural feature discovery at different levels and enhancing the understanding of network structure (*Ma et al., 2021*). *Khawaja et al. (2021)* put forward a method for detecting implicit or weak communities in a network by attenuating the strength of the main structure. The test results indicated a substantial disparity in the number of identified communities compared to conventional algorithms (*Khawaja et al., 2021*). Overall, these advances demonstrate the ongoing efforts to develop more accurate and effective methods for structure exploration in various applications.

Network structure exploration requires adapting to more networks and discovering multiple structural features to gain more information about complex systems. However, there are still challenges at present. Many algorithms have over-designed models, and there is limited research on structural characteristics beyond community structure (*Cantwell & Newman, 2019*; *Fei et al., 2023*). Using deep learning for network structure exploration is a feasible approach that has achieved some results (*Pham et al., 2022*; *Shun et al., 2022*). However, interpretability remains a challenge.

Therefore, this article proposes a statistical inference method based on the flexible generative model to explore network structures. The latent parameters of the model are calculated using the belief propagation algorithm (*Qu, Tang & Yan, 2021*) based on Markov random fields. A parameter configuration scheme was proposed to improve the convergence speed and performance by combining the parameter initialization experience of deep learning models. A fusion algorithm strategy was proposed to explore the multi-type structural features and improve the adaptability of the algorithm.

## MATERIALS AND METHODS

For the network $G$, we assign a latent parameter $x$ to each node. Considering a pair of nodes $i$ and $j$, the probability of an edge between them can be expressed by Eq. (1).

$$p_{ij} = \frac{s_i s_j}{2l} f(x_i, x_j) \tag{1}$$

In Eq. (1), $s_i$ represents the strength of node $i$, which is the sum of the edge weights connected to it. $l$ represents the total strength of the network, which is the sum of the weights of all edges. $f(x_i, x_j)$ is the probability measure that maps a pair of the latent parameters to the range $(0, 1)$.

We used the binary function form of the generalized Bernstein polynomials to approximate the probability measure. It can be expressed as Eq. (2).

$$f(x_i, x_j) = \sum_{u,v=0}^{M} \beta_{uv} C_M^u x_i^{u\alpha} \left(1 - x_i^{\alpha}\right)^{M-u} C_M^v x_j^{v\alpha} \left(1 - x_j^{\alpha}\right)^{M-v} \tag{2}$$

In Eq. (2), $M$ represents the order of the Bernstein polynomial, and $\beta_{uv}$ represents a set of model parameters that need to be initialized and iteratively updated during computation. The $\alpha$ is a hyper parameter, which is set to 1 in the experiments to simplify calculations. $C_M^u$ and $C_M^v$ represent the combination coefficients of the Bernstein polynomial.

By using the maximum likelihood estimation method, we can iteratively calculate the values of hidden parameters. The distribution of these values can reflect the structure features of the network. In order to improve the adaptability of the algorithm, we consider combining it with the label propagation algorithm. The label propagation algorithm is represented by Eq. (3).

$$L_i = \arg\max_{L} \sum_{j \in \partial(i)} \delta(L_j, L) \cdot g(d_j) \tag{3}$$

In Eq. (3), $L_i$ represents the label of the target node i, while $L_j$ denotes the label of the neighbor node j. $\partial(i)$ represents the set of neighbors of the target node i, and $\delta(L_j, L)$ is the Kronecker function. $g(d_j)$ is a quantization function. $d_j$ represents the degree of the neighbor node j.

Usually, model parameters can be randomly initialized, but in the experiments, we found that the effect of random initialization is very poor. To address this challenge, we propose an initialization configuration scheme for the model parameters $\beta_{uv}$, which is formulated as Eq. (4). This configuration scheme is a feasible solution obtained after conducting numerous experiments.

$$\beta_{uv} = \begin{cases} e^{u-v}, & u < v \\ \bar{d} - \left\lfloor \dfrac{\bar{d}}{10} \right\rfloor \times 10 + 1, & u = v \\ \beta_{vu}, & u > v \end{cases} \tag{4}$$

The initialization configuration scheme for the model parameters $\beta_{uv}$ primarily consists of two parts. The values of the diagonal elements in the parameter matrix are initialized within the range of $[1, 11)$, while the values of the remaining elements lie within the interval $(0, 1)$. The parameter matrix is symmetric because we are studying undirected networks. This configuration approach increases the values of the diagonal elements to ensure the salience of features. The off-diagonal elements are mapped to dispersed values using an exponential function to maintain generalization ability. In the subsequent experimental section, we compared our configuration scheme with several typical schemes to demonstrate its effectiveness.

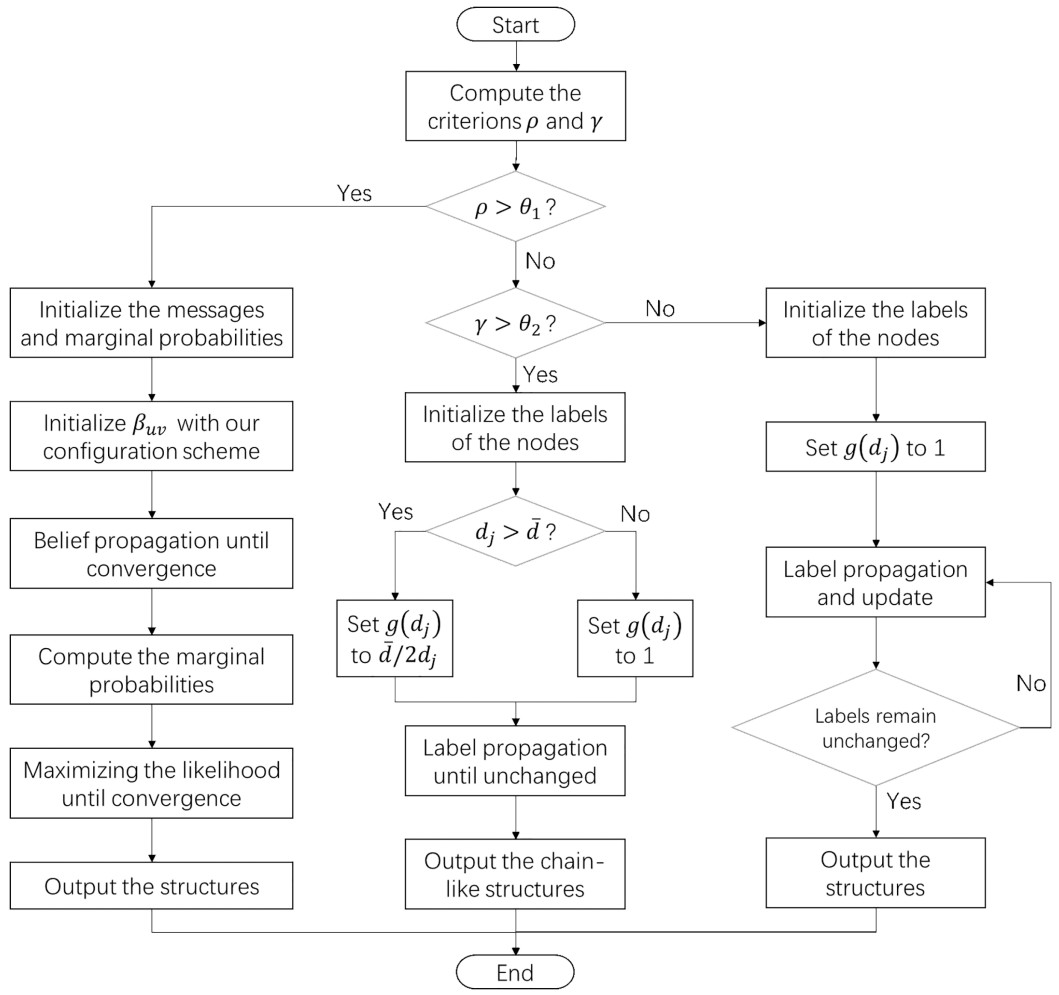

**Figure 1 The flowchart of the structure exploration algorithm for complex systems.** When the sparsity of the target network is higher than the threshold $\theta_1$, the algorithm will use the generative model and our configuration scheme to discover the composite structures. Otherwise, the algorithm detects chain-like structures or cluster structures in the network according to the threshold $\theta_2$.

The algorithm takes a long time on some sparse networks, so we propose an algorithm fusion strategy. The fusion criterion is the sparsity of the target network, which is calculated using Eq. (5).

$$\rho = \frac{2m}{N(N-1)} \tag{5}$$

We introduce hyper parameter $\theta_1$ in the fusion algorithm to control the switching of the algorithm flow. The adaptability of the algorithm has been improved. However, we found that some applications also benefit from chain-like structures, such as route planning and supply chain analysis. Therefore, we propose the criterion $\gamma$, which

represents the proportion of nodes with a degree of 2 in the total number of nodes, as defined by Eq. (6).

$$\gamma = \frac{1}{N} \sum_{i=1}^{N} \delta(d_i, 2) \tag{6}$$

We introduce hyper parameter $\theta_2$ to determine when to detect chain-like structures. The detection of chain-like structures can be achieved by adjusting the $g(d_j)$ function, as specified in Eq. (7).

$$g(d_j) = \begin{cases} \bar{d}/2d_j, & d_j > \bar{d} \\ 1, & d_j \le \bar{d} \end{cases} \tag{7}$$

By integrating the configuration scheme and the algorithm fusion strategy, we have proposed a network structure exploration framework with high adaptability. This framework does not require prior knowledge and is able to discover various structural features. The flowchart of this algorithm is illustrated in Fig. 1.

## RESULTS

The experiment consists of two main parts. The first part involves comparing the model parameter configuration schemes. The second part focuses on testing the performance of the algorithm. Through these experiments, we are able to confirm the effectiveness of our model parameter configuration scheme and demonstrate the performance of our algorithm.

We utilized the program developed by *Lancichinetti & Fortunato (2009)* to generate a series of benchmark networks. These generated benchmark networks all contain community structures. The details of these benchmark networks can be found in the data provided in Table 1.

We conducted experiments on the benchmark networks, comparing our scheme with four representative configuration schemes. The schemes are as follows:

Random: Parameters are randomly initialized.

Ones: All initial parameters are set to 1.

Ones-Random: The diagonal of the parameter matrix is set to 1, while the rest are random numbers.

Random-Ones: The diagonal of the parameter matrix is set to random numbers, while the rest are 1.

Configuration: Refers to the model parameter configuration scheme mentioned earlier in this document.

The last three configuration schemes separate the parameters for diagonal and off-diagonal elements. Because the model is constructed to fit the adjacency matrix of the target network. The diagonal elements of the adjacency matrix represent self-connections, which differ from the off-diagonal elements.

**Table 1 The information of the benchmark networks.**

| No. | Nodes | Edges | Average degree | Sparsity |
|---|---|---|---|---|
| B1 | 1,000 | 1,030 | 2.06 | 0.002062 |
| B2 | 1,000 | 1,110 | 2.22 | 0.002222 |
| B3 | 1,000 | 1,602 | 3.204 | 0.003207 |
| B4 | 1,000 | 1,747 | 3.494 | 0.003497 |
| B5 | 1,000 | 2,256 | 4.512 | 0.004517 |
| B6 | 1,000 | 2,615 | 5.23 | 0.005235 |
| B7 | 1,000 | 3,024 | 6.048 | 0.006054 |
| B8 | 1,000 | 4,357 | 8.714 | 0.008723 |
| B9 | 1,000 | 5,938 | 11.876 | 0.011888 |
| B10 | 1,000 | 7,547 | 15.094 | 0.015109 |

**Table 2 The pass rates of the schemes.**

| Scheme | Random | Ones | Ones-random | Random-ones | Configuration |
|---|---|---|---|---|---|
| Pass rate | 60.00% | 0.00% | 20.00% | 0.00% | 100.00% |

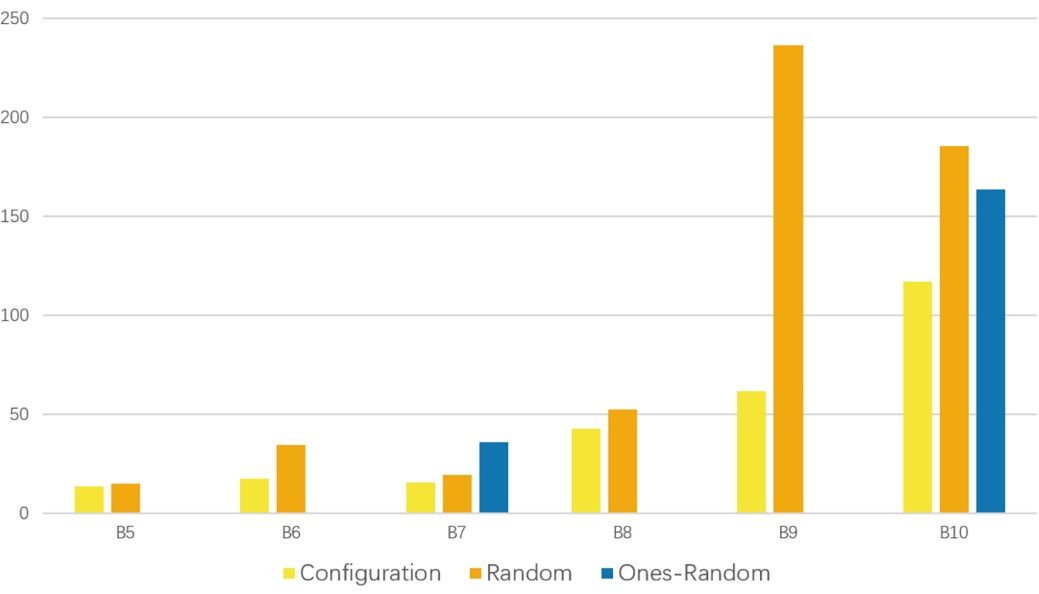

**Figure 2 Comparison of the time consumption for schemes passing the tests.** Each time consumption represents the average performance of 10 runs. The yellow bars represent the time consumption of our configuration scheme. It is shorter in each group of experiments that have passed the tests.

Then the statistical inference algorithm based on belief propagation was employed for community detection. The experimental results can be reflected by the standard deviation of the hidden node parameters in the network model. Since the benchmark networks contain community structures, when the standard deviation of the detection results is

**Table 3** The information of the two real-world road networks.

| No. | Nodes | Edges | Average degree | Sparsity | γ |
|---|---|---|---|---|---|
| RN2019 | 156 | 167 | 2.141 | 0.013813 | 82.692% |
| RN2020 | 181 | 245 | 2.707 | 0.015040 | 58.564% |

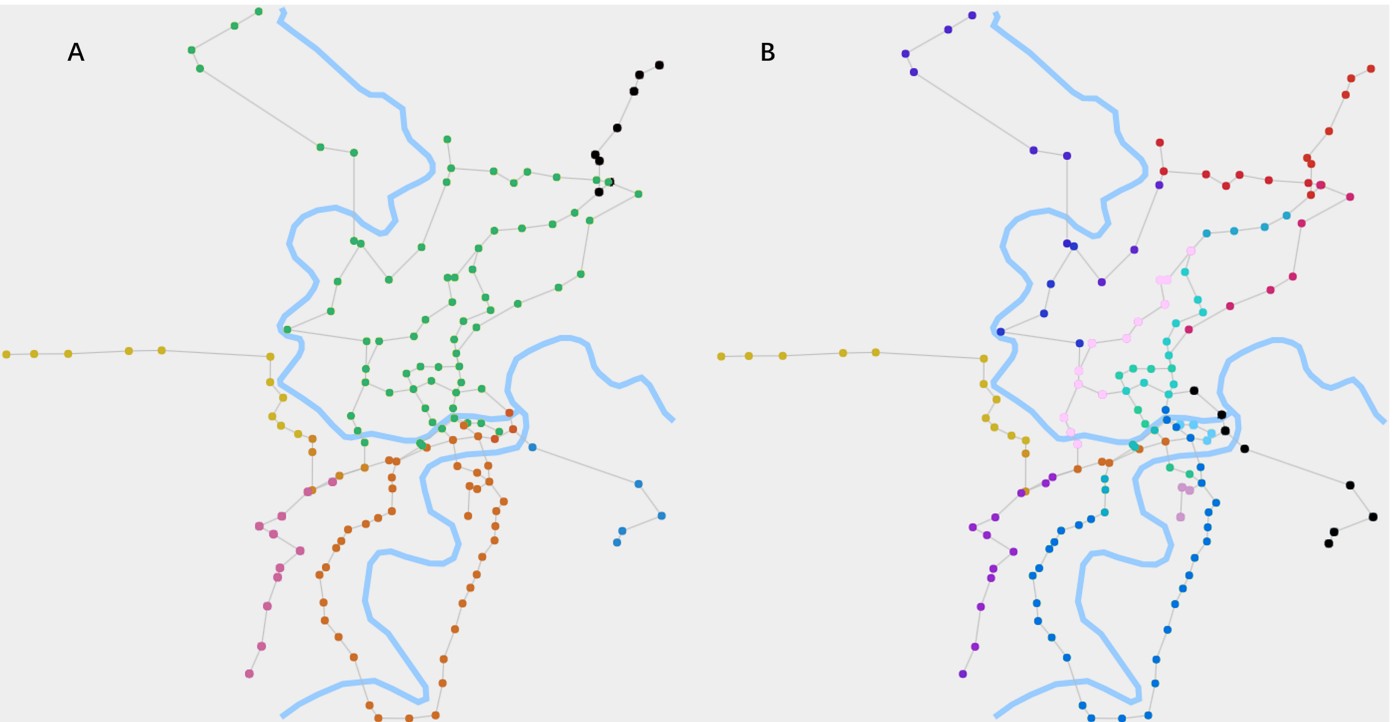

**Figure 3 The cluster structures and chain-like structures of RN2019 detected by our algorithm.** (A) When $\theta_1$ is 0.015 and $\theta_2$ is 0.85, the algorithm detects the cluster structures. It reflects the zoning of the city's road network by the node colors. (B) When $\theta_1$ is 0.015 and $\theta_2$ is 0.6, the algorithm detects the chain-like structures. It reflects the routes of the road network by the node colors.

particularly small, it indicates that the communities have not been detected. Therefore, the corresponding scheme exhibited a failure.

Based on the experimental results presented in Table 2, we can clearly observe that the Random scheme passed 6 out of the 10 benchmark networks tested, while the Ones-Random scheme passed two groups. The Ones and Random-Ones schemes failed to pass any of the tests. In contrast, our model configuration scheme passed all tests. This indicates that our configuration scheme has effectively improved the adaptability of the algorithm.

Figure 2 presents the recorded detection time, which indicates that our scheme exhibits a notable advantage in terms of speed.

Next, we conducted tests to evaluate the performance of the fusion algorithm. To provide a more intuitive demonstration, we prepared two real-world road networks. These road networks were collected in 2019 and 2020, respectively. The nodes represent stations

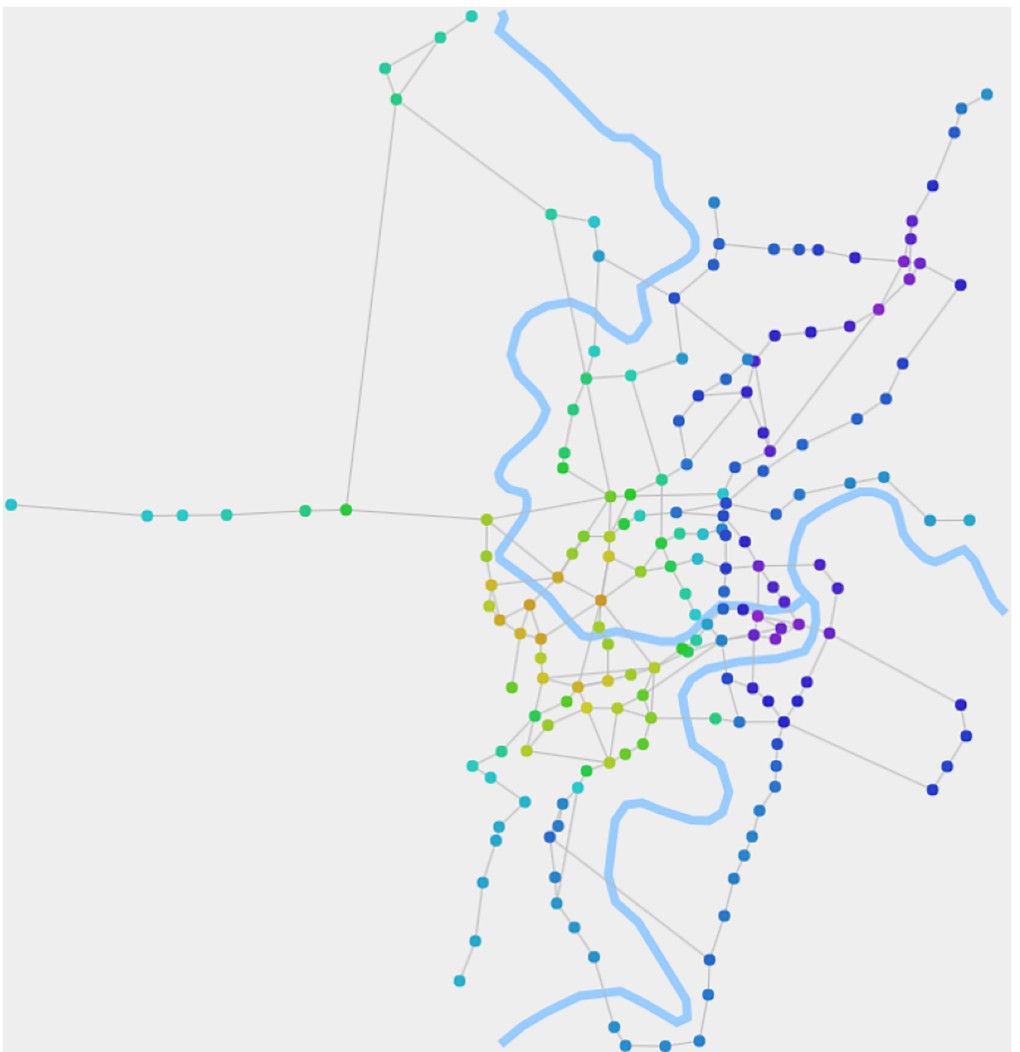

**Figure 4 The composite structures of RN2020 detected by our algorithm.** When $\theta_1$ is 0.015, the algorithm detects the composite structures. It depicts a more detailed internal structure through the gradient and transition of node colors.               

in a city and the edges represent roads between stations. The specific information of these road networks is presented in Table 3.

The two hyper parameters, $\theta_1$ and $\theta_2$, in the fusion algorithm can be adjusted as needed. In this experiment, after calculating the relevant indicators of the networks, we set $\theta_1$ to 0.015 and $\theta_2$ to 0.85 for the first experiment. Then, keeping $\theta_1$ constant, we adjusted $\theta_2$ to 0.6 for the second experiment.

To visualize the detection results of the fusion algorithm, we embedded them into a map, where the internal structure is represented by node color. Figure 3 illustrates this representation.

In Fig. 3A, the cluster structures of RN2019 are clearly manifested, which are computed using the label propagation process of the fusion algorithm. It broadly reflects the zoning of the city's road network. Figure 3B shows the chain-like structures of RN2019, which are

obtained through the process of inhibiting competitiveness of high-degree nodes. It clearly represents the routes of the road network.

Figure 4 displays the composite structures of RN2020, which are calculated using the belief propagation process of the fusion algorithm. It not only reflects the zoning of the city's road network but also depicts a more detailed internal structure through the gradient and transition of node colors.

## DISCUSSION

Unlike previous community detection research, this method not only explores cluster structures in the network but also detects chain-like, overlapping, and other structures based on network characteristics. This network structure exploration capability makes it more adaptable and applicable.

This method does not require excessive prior knowledge about the target network, which is convenient for preliminary analysis of complex systems. The implementation of model configuration scheme and algorithm fusion strategy in this method is simple, leading to improved adaptability and convergence speed.

The configuration of model parameters may seem unimportant, but it actually has a significant impact on the execution process and results of the algorithm. Therefore, after conducting numerous experiments, this article has summarized a feasible configuration scheme. These efforts have made the use of the algorithm more convenient and enabled its quick application.

## CONCLUSIONS

To support the analysis and research of complex systems, this article proposes an algorithm for exploring network structures that combines model configuration and algorithm fusion. The algorithm is capable of exploring various structural features within a network based on network indicators. It demonstrates good stability and adaptability.

Furthermore, experiments are provided in this article to demonstrate the effectiveness of the algorithm. Some applications of the algorithm are demonstrated, which can serve as a reference for cross-disciplinary research in related fields.

## ACKNOWLEDGEMENTS

I would like to express my gratitude to Dr. Lianggui Tang for his guidance on the general research direction of this article, and to Dr. Ban Wang for his assistance in selecting the journal.

### Funding

This work was supported by the Science and Technology Research Program of Chongqing Municipal Education Commission (Nos. KJZD-M202200801, KJQN202200828 and KJQN201800807). The funders had no role in study design, data collection and analysis, decision to publish, or preparation of the manuscript.

## Grant Disclosures

The following grant information was disclosed by the authors:
Science and Technology Research Program of Chongqing Municipal Education
Commission: KJZD-M202200801, KJQN202200828, KJQN201800807.

## Competing Interests

Wanbing Liu is employed by Hengda Fuji Elevator Co. Ltd.

## Author Contributions

- Yingfei Qu conceived and designed the experiments, performed the experiments, analyzed the data, performed the computation work, prepared figures and/or tables, authored or reviewed drafts of the article, and approved the final draft.
- Wanbing Liu analyzed the data, authored or reviewed drafts of the article, and approved the final draft.
- Junhao Wen conceived and designed the experiments, analyzed the data, performed the computation work, authored or reviewed drafts of the article, and approved the final draft.
- Ming Li performed the experiments, prepared figures and/or tables, and approved the final draft.

## Data Availability

The Java project files for the code and the data are available in the Supplemental File.

## Supplemental Information

Supplemental information for this article can be found online at http://dx.doi.org/10.7717/peerj-cs.1983#supplemental-information.

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
