# Peer review of "Adaptive robust structure exploration for complex systems based on model configuration and fusion"

_PeerJ Computer Science, doi:10.7717/peerj-cs.1983_

## Round 0.1 · original submission · Major Revisions

I recommend Major Revision for the manuscript.
Reviewer 2, who recommended rejection gave as observations only generalities.

**Language Note:** The review process has identified that the English language must be improved. PeerJ can provide language editing services - please contact us at [email protected] for pricing (be sure to provide your manuscript number and title). Alternatively, you should make your own arrangements to improve the language quality and provide details in your response letter. – PeerJ Staff

Reviewer 1 ·

Basic reporting

Reviewed on:Feb 12, 2024
Journal: PeerJ Computer Science
Review: Major Changes


Adaptive robust structure exploration for complex systems based on model configuration and fusion

This technical article, “Adaptive robust structure exploration for complex systems based on model configuration and fusion,” is a novel approach. An adaptive algorithm designed to explore intricate system structures employs a generative model. Through a fusion of model configuration and algorithm, it enhances robustness and convergence speed. By analyzing and fitting potential structures within complex network systems, the algorithm uncovers diverse structural characteristics such as clustering, overlapping, and local chaining. This thorough examination yields valuable insights for understanding system functions and facilitating in-depth research on control mechanisms. In summary, this adaptive algorithm presents a promising methodology for exploring complex system structures, offering a valuable tool for comprehensive analysis across multiple fields by capturing a range of structural features beyond traditional approaches.

Please elaborate further on the Ones-Random: The diagonal of the parameter matrix is set to 1, while the rest are random 186 numbers. The rationale behind the numbers.
This paper requires significant grammatical improvements.

Please make sure to incorporate these changes.

Experimental design

The initialization configuration scheme for the model parameters primarily consists of two parts. The values of the diagonal elements in the parameter matrix are initialized within the range of 1,11. Could you elaborate on why these numbers were chosen?

Validity of the findings

Please elaborate further on the Ones-Random: The diagonal of the parameter matrix is set to 1, while the rest are random 186 numbers. The rationale behind the numbers.
This paper requires significant grammatical improvements.

Reviewer 2 ·

Basic reporting

The Paper "Adaptive robust structure exploration for complex systems based on model configuration and fusion" presents an adaptive algorithm to explore complex systems. The manuscript lacks in the following aspects.
1- The abstract falls short in providing essential details regarding the novelty of the research, the techniques employed to ensure robustness in exploring the complex system, and a clear articulation of the field's problem statement. Additionally, the results obtained from the study are conspicuously absent from the abstract. Enhancing the abstract to incorporate these crucial elements will offer readers a more comprehensive understanding of the research's significance and contributions to the field.
2-The introduction should orient the reader to the problems and research gaps within the field by thoroughly exploring the state-of-the-art and recent literature. However, the authors exhibit inconsistency in this section, as they fail to provide a cohesive and comprehensive overview of existing knowledge, leaving gaps in the reader's understanding of the context and the specific issues the research aims to address.

Experimental design

The manuscript needs a thorough overhaul. Writing is not coherent. Results and experimentation are not properly explained.

Validity of the findings

The results section can be improved further by adding more discussion on results.
The captions of figures and tables are not properly explained.

Additional comments

The manuscript can be improved in the light of above comments.

Reviewer 3 ·

Basic reporting

The abstract section is fragile. Please re-write an abstract section, explain an obtained result and contribution, improve a proposed method, etc.

How was the generative model utilized in the algorithm to explore complex system structures, and what advantages does it offer compared to traditional approaches?

How were the 173 model parameter configuration schemes identified and selected for comparison in the experiment, and were there any specific criteria used for their inclusion?

Did the literature review encompass a comprehensive examination of existing approaches to model parameter configuration in complex network analysis, and how did it inform the design of the experimental setup?

What motivated the need to explore and identify the optimal parameter configuration for the model, and how does it contribute to addressing the research problem at hand?

Experimental design

What specific procedures were followed to conduct the comparative experiments on the benchmark networks, and how were the five different configuration schemes implemented and evaluated?

How was the statistical inference algorithm based on belief propagation employed for community detection, and what considerations were taken into account to ensure the validity and reliability of the results?

What were the key findings from the comparative experiments on the benchmark networks, particularly in terms of the performance and effectiveness of the different configuration schemes?

Validity of the findings

Please elaborate in detail on how the experimental results demonstrate the superiority of the proposed model configuration scheme over alternative approaches, and what insights were gained from analyzing the standard deviation of the hidden node parameters.

There is a need to discuss how the experiments' results were evaluated to ensure their robustness and generalizability, particularly in terms of the reliability of the statistical inference algorithm and the validity of the detection results.

Were there any limitations or potential biases in the experimental setup that could have influenced the interpretation of the results, and how were they addressed to enhance the credibility of the findings?

Additional comments

The introduction should provide more background on the project with the scope of the work and motivations.
The paper, does not link well with recent literature on top-tier journals research gap should be clearly identified.

---

## Round 0.2 · accepted · Accept

Accept the paper to be published.

Reviewer 1 ·

Basic reporting

Based on my previous review, it seems like the author has cooperated with the comments, and I'm okay with the changes they made

Experimental design

The experimental design is okay and they have used binary function form of the generalized Bernstein polynomials to approximate the probability measure.

Validity of the findings

In the findings, parameters are split between diagonal and off-diagonal elements to match the target network's adjacency matrix. This design accounts for self-connections represented by diagonal elements distinct from off-diagonal connections, which I consider a good approach.

Reviewer 2 ·

Basic reporting

The authors have significantly improved the manuscript before acceptance authors are advised to make some minor changes to improve the manuscript's readability.
1- To ensure consistency in referring to figures in the text, it's essential to maintain uniformity. You can choose either "Fig." or "Figure" and use it consistently throughout the text.
2- Mention the x and y axes of Figure 2.
3-Explaian all the metrics mentioned in Tables 1 and 2. What is the average degree mentioned in Table 1 and how is it calculated?

Experimental design

no comment

Validity of the findings

The description mentioned in conclusion seems very general. Authors are advised to add some details about their proposed method and attained results based on used metrics.